

# Description of the microbiota in epidermal mucus and skin of sharks (*Ginglymostoma cirratum* and *Negaprion brevirostris*) and one stingray (*Hypanus americanus*)

Susana Caballero[1,*], Ana Maria Galeano[1,*], Juan Diego Lozano[1] and Martha Vives[2]

[1] Laboratorio de Ecología Molecular de Vertebrados Acuáticos, LEMVA, Biological Sciences Department, Universidad de los Andes, Bogota, Colombia
[2] Centro de Investigaciones Microbiológicas, CIMIC, Biological Sciences Department, Universidad de los Andes, Bogota, Colombia
[*] These authors contributed equally to this work.

Corresponding author
Susana Caballero,
sj.caballero26@uniandes.edu.co

## ABSTRACT

Skin mucus in fish is the first barrier between the organism and the environment but the role of skin mucus in protecting fish against pathogens is not well understood. During copulation in sharks, the male bites the female generating wounds, which are then highly likely to become infected by opportunistic bacteria from the water or from the male shark's mouth. Describing the microbial component of epithelial mucus may allow future understanding of this first line of defense in sharks. In this study, we analyzed mucus and skin samples obtained from 19 individuals of two shark species and a stingray: the nurse shark (*Ginglymostoma cirratum*), the lemon shark (*Negaprion brevirostris*) and the southern stingray (*Hypanus americanus*). Total DNA was extracted from all samples, and the bacterial 16S rRNA gene (region V3-V4) was amplified and sequenced on the Ion Torrent Platform. Bacterial diversity (order) was higher in skin and mucus than in water. Order composition was more similar between the two shark species. Alpha-diversities (Shannon and Simpson) for OTUs (clusters of sequences defined by a 97% identity threshold for the16S rRNA gene) were high and there were non-significant differences between elasmobranch species or types of samples. We found orders of potentially pathogenic bacteria in water samples collected from the area where the animals were found, such as Pasteurellales (i.e., genus *Pasteurella* spp. and *Haemophilus* spp.) and Oceanospirillales (i.e., genus *Halomonas* spp.) but these were not found in the skin or mucus samples from any species. Some bacterial orders, such as Flavobacteriales, Vibrionales (i.e., genus *Pseudoalteromonas*), Lactobacillales and Bacillales were found only in mucus and skin samples. However, in a co-occurrence analyses, no significant relationship was found among these orders (strength less than 0.6, p-value > 0.01) but significant relationships were found among the order Trembayales, Fusobacteriales, and some previously described marine environmental Bacteria and Archaea, including Elusimicrobiales, Thermoproteales, Deinococcales and Desulfarculales. This is the first study focusing on elasmobranch microbial communities. The functional role and the benefits of these bacteria still needs

understanding as well as the potential changes to microbial communities as a result of changing environmental conditions.

# INTRODUCTION

The first barrier of protection against microorganisms in fish is the mucosal immune system (*Cone, 2009*). This system protects fish physically, chemically, and biologically from threats or pathogens found in their habitat (*Subramanian, MacKinnon & Ross, 2007*; *Subramanian, Ross & MacKinnon, 2008*; *Raj et al., 2011*). The mucosal immune system is subdivided into three subgroups that correspond to the locations where the mucus is secreted: the gut, the gills and the skin (*Salinas, Zhang & Sunyer, 2011*). Some studies suggest that this mucus is constantly renewed, reducing the pathogenic load found on the surface of the fish (*Nagashima et al., 2003*). The mucus is secreted in higher quantities as a response to threat (*Mittal & Datta Munshi, 1974*; *Gostin, Neagu & Vulpe, 2011*; *Rai et al., 2012*), and this viscous substance consists of molecules that may help in healing and protecting the skin (*Cameron & Endean, 1973*; *Al-Hassan et al., 1985*), including the secretion of antimicrobial and regenerative compounds (*Hansen & Olafsen, 1999*).

The epithelial mucus is sometimes considered an ideal surface for bacterial adhesion. In fact, the accumulation of microorganisms appears to take place during the lifetime of the individual (*Hansen & Olafsen, 1999*), leading to the establishment of the microbiota in fish skin. However, it is also recognized that the mucus has a concentration of molecules that prevent the adhesion of pathogenic bacteria (*Crouse-Eisnor, Cone & Odense, 1985*). As such, the role or the relationship between the mucus and environmental bacteria is not clear (*Luer, 2012*). It has been suggested that bacteria found in this layer may play three possible roles (*Salminen et al., 2010*): (a) bacteria may stimulate mucus and antimicrobial compound production, (b) bacteria may activate and help modulate the immune response in the fish, and (c) the interaction between different types of bacteria may actively exclude or compete with potentially pathogenic bacteria.

The mucus layer in sharks and rays has been poorly studied. However, it is known that mucus from stingray skin appear to accelerate the healing processes of wounds, and that bacteria found in the mucus present antibacterial activity against human pathogens (*Luer et al., in press*). Also, it has been found recently that the structure, geometry and arrangement of dermal denticles of the shark skin play an important role in allowing bacterial attachment and development of biofilms (*Chien et al., 2020*). Reproductive behavior in this group is characterized by aggressiveness during courtship and copulation (*Pratt & Carrier, 2001*; *Carrier, Pratt & Martin, 2015*). In sharks, the male bites the female on her dorsal or pectoral fins generating wounds in those areas (*Pratt & Carrier, 2001*). Polyandry, a mating system in which one female mates with multiple males, is very common in some species (*Saville et al., 2002*; *Carrier et al., 2003*). This behavior drives

competition between males and avoidance in females (*Klimley, 1980*; *Gordon, 1993*; *Pratt & Carrier, 2001*). There are also morphological characteristics related to this trait. Sexual dimorphism occurs in shark species in which the male's teeth are shaped so they can easily grab the female in order to remain close to her while mating. Females have thicker dermal denticles (tooth-like structures that provide hydrodynamics and protection) than males as protection against these bites (*Carrier, Musick & Heithaus, 2012*). In the case of rays, the females prick the male with their caudal spine (*Pratt & Carrier, 2001*). It has been shown in some stingray species that when many males are involved in mating, a few may die in the process (*Gilad et al., 2008*). In spite of these apparently aggressive behaviors, copulation is necessary and the wounds provoked are highly likely to become infected (*Daly-Engel et al., 2010*) due to opportunistic bacteria in the water and in the oral cavity of males. Because of the high concentration of pathogenic microorganisms found in the aquatic environment (*Magnadottir, 2010*), it is important to determine the microbiota component of the epithelial mucus, the skin, and to understand whether the bacteria found in these are similar or different from those found in the water surrounding the animals. This will help to understand the role of mucus in the protection against pathogens. In this study, we characterized the bacterial diversity in the epithelial mucus in three elasmobranch species, the nurse shark (*Ginglymostoma cirratum),* the lemon shark (*Negaprion brevirostris)* and the southern stingray (*Hypanus americanus)* (*Last, Naylor & Manjaji-Matsumoto, 2016*). We also hypothesize about the possible role of some of the bacteria found in the mucus and in the skin.

## MATERIALS AND METHODS

### Sample collection

Mucus and skin tissue samples were collected from 19 apparently healthy individuals (no visible wounds, normal swimming activity); 14 of them from animals captured in Bimini, Bahamas ($25°43'59N$, $79°14'60W$): four corresponded to juvenile nurse sharks (*Ginglymostoma cirratum)*, six to juvenile lemon sharks (*Negaprion brevirostris),* and four to adult southern stingrays (*Hypanus americanus)*. Samples from an additional five adult nurse sharks were collected at Oceanario from Islas del Rosario (CEINER), in the Colombian Caribbean ($10°10'30N$, $75°45'00W$). For each individual, we obtained a sample of skin tissue and mucus, following sampling protocols approved by the Animal Care Committee of Universidad de los Andes (CICUAL) (Bogota, Colombia). The skin tissue sample was cut, using a sterile blade for each specimen, from the posterior part of the dorsal fin ($1 \text{ cm}^3$ or less) and the mucus from the skin surface, using a sterile 1.5 ml microcentrifuge tube to scrape the skin surface, ideally filling at least half of the tube. Animals were manipulated for approximately 5 min and immediately released. A water sample was also collected in sterile 15 ml tube from the sampling location of each individual. Thus, three samples were associated with each individual, for a total of 57 samples. The individuals were captured and raised slightly above the surface of the water, so that the samples could be taken outside the water, while the animal could continue breathing. Skin samples were preserved in ethanol 90%. All samples were maintained at 4 °C for less than one week, until processing.

## DNA Extraction and PCR amplification

DNA was extracted from the entire sample collected for all samples. The Tissue and Cells DNA Isolation Kit (MoBio Laboratories, Inc.) was used, following the manufacturer instructions. Water samples were filtered through a 0.8 μm cellulose nitrate filter before DNA extraction. The primers 515f and 806r were used in order to amplify the region V4 from the bacterial and archaea 16S rRNA gene using the primers 515F (5′-GTGCCAGCMGCCGCGGTAA-3′) and 806R (5′GGACTAHVGGGTWTCTAAT-3′) (*Caporaso et al., 2010*). PCR amplification conditions were as follows: an initial denaturation at 94 °C for 3 min, followed by 35 cycles of denaturing at 94 °C for 45 s, annealing for 45 s at 50 °C and extension for 45 s at 72 °C, followed by a final extension of 20 min at 72 °C. A negative PCR control was always included to reduce the chance of contaminant amplification. Successful amplification was confirmed on 1% agarose gel.

## Ion torrent library preparation, quantification and sequencing

From the 57 samples, 32 were used to construct libraries (Table S1). Samples were chosen depending on their final DNA concentration, once the PCR products were cleaned using magnetic beads and run on a 1.5% agarose gel. Only the samples that had a clear strong band were used for library construction. Two libraries, each with 16 barcodes, were prepared using the protocol Ion Xpress™ Plus gDNA Fragment Library Preparation (Life Technologies). Libraries were quantified with the Qubit kit. Templates were prepared following the Ion PGM™ Template OT2 200 Kit (Life Technologies) protocols. Libraries were prepared for sequencing using the protocol Ion PGM™ Sequencing 200 Kit v2 (Life Technologies). Libraries were pooled to equimolar concentration and loaded on two Ion 316 chips and sequenced in the Ion Torrent PGM (Life Technologies).

16S rRNA datasets used in this manuscript with accompanying metadata has been submitted to Dryad as https://datadryad.org/stash/dataset/doi:10.5061/dryad.b5mkkwh8j.

## Bioinformatic and statistical analyses

Sequences were separated by barcodes directly by the Ion Torrent PGM and saved by the ion reporter in different files; sequence quality was analyzed using FastQC (*Andrews, 2014*). The file format was changed from Fastq to Fasta. Demultiplexing was conducted by comparing the mapping file of the chip with the files containing the sequences. For the core diversity analysis, Qiime2 (*Bolyen et al., 2019*) was used via command line using the moving pictures tutorial as reference. The files were imported as "MultiplexedSingleEndBarcodeInSequence" and demultiplexed using "cutadapt", eliminating sequences shorter than 50 bp. The sequences went through DADA2 (*Callahan et al., 2016*) for quality control to delete sequences with lower qscore than 20 and then the remaining sequences were aligned *de novo* with align-to-tree-mafft-fastree. In parallel, the sequences were clustered into OTUs used to perform non phylogenetic analysis. The rooted tree obtained with fasttree2 (*Price, Dehal & Arkin, 2010*) was used to perform an alpha rarefaction with a 1,000 sequence depth. For taxonomic assignment, analyses were performed on the Galaxy online platform (*Afgan et al., 2016*) following one amplicon data workflow on Mothur v.1.28.0 (*Schloss et al., 2009*). This workflow started by merging all

read files into group files. Group files were identified as samples from each of the three elasmobranch species and also as type of sample (skin, mucus or water). The next step of the workflow identified unique sequences and generated a file with these sequences and a second file in which the number of each unique representative sequence was kept. Following this, reads were filtered based on quality and length. Parameters to remove low quality sequences (quality control) was for those with less than 20 Phred score and shorter than 50 bp. (minimum length) followed by a step to remove poorly aligned sequences and chimeric sequences. Finally, reads were clustered based on their degree of similarity, with a minimum of 97% identity threshold and aligned to the Silva V4 reference database (*Quast et al., 2013*), followed by a classification step into taxonomic categories (order, family, genus and species when possible).

Rstudio version 1.1.463 (*R Development Core Team, 2010*) was used (*Wickham, 2009*) for alpha ($\alpha$) diversity analyses (Simpson and Shannon) (package Vegan) (*Oksanen et al., 2015*) which were conducted for OTUs, using the number of OTUs per sample, comparing among species (*N. brevirostris*, *G. cirratum* juveniles and adults and *H. americanus*) and among sample types (tissue, mucus and water). In this analysis, OTUs with less than 0.2% presence were not included. A Shapiro–Wilk normality test was conducted to evaluate normality among samples belonging to each elasmobranch species (Table S1), including the additional category of adults and juvenile for nurse sharks, or to each category of sample type before performing any statistical tests. Since results fell outside the normality assumption, a Kruskal–Wallis test was used to evaluate whether $\alpha$ diversity was significantly different among elasmobranch species or among sample type. To estimate beta ($\beta$) diversity (Bray–Curtis dissimilarity index and a Principal Component Analysis (PCA) the taxonomic category "order" was used. Venn diagrams (package DVenn) were used to visualize shared orders among elasmobranch species and among sample types.

In order to find co-occurrence between different bacterial and/or Archaea orders a correlation matrix was created in R using the Spearman's co-efficient as in *Ju et al. (2013)*. Correlations had to be stronger than 0.6 with a *p*-value < 0.01 to be considered to have a significant co-occurrence with other orders. All orders, including those with less than 0.2% presence were included in the co-occurrence analysis. A chord plot was created to visualize the relations between the different orders.

## RESULTS

The 32 samples used to build the libraries included a mucus or tissue sample for each of the individuals sampled and only four of the 19 samples from water (Table S1). The other samples, including 15 water samples, had low DNA concentrations that could not be used for NGS sequencing analysis, characterized by weak or no bands amplified. A total of 219,162 reads were obtained from the Ion Torrent PGM of which 55,642 were used for subsequent analyses following demultiplexing. After read quality control and chimera removal, 21,530 reads were used in the following steps for Qiime2 and in the Mothur workflow. Of these, 17,685 were grouped as unique OTUs in Qiime2 (most of them represented each by only one read, Fig. S1) (82% total reads). In Mothur, 3,639

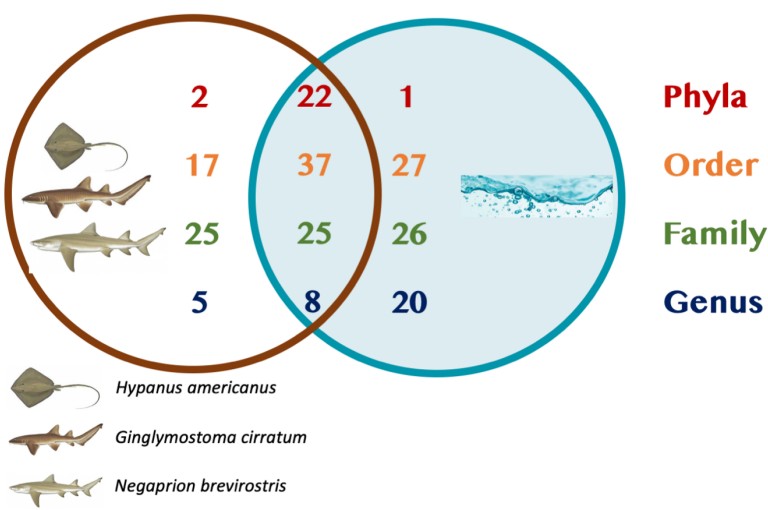

**Figure 1** **Summary of taxonomic assignments.** Venn diagram showing the number of orders shared among elasmobranch samples (mucus and tissue), and water samples. Also, orders unique to either elasmobranch samples (brown circle) or water samples (blue circle).

(16.9% total reads) were assigned taxonomically against the SilvaV4 database while 19,164 were left unassigned (84% total reads); sequences assigned taxonomically belonged to 18 mucus, 10 skin and 4 water samples.

A total of 25 phyla, 81 orders, 76 families and 33 genera were assigned (Fig. 1), but analyses were restricted to OTUs and to the taxonomic category "order", since this was the category with higher levels of taxonomic assignments. Most reads were identified as belonging to the Bacteria Domain (Table S2). Occurrence of reads belonging to the kingdom Archaea was low (<2%) and these were only found in one mucus sample from one nurse shark. Among the Archaea Domain, the orders identified were Micrarchaeles, Cenarchaeales, Halobacteriales, and Methanobacteriales. Thirty-seven orders were shared between samples from the three elasmobranch species and the water samples, and 17 were solely found in elasmobranch samples. Twenty-seven orders were found only in the water samples (Fig. 1). Fifty-four were shared among all elasmobranch species, 28 were shared between the nurse shark and the lemon shark, and 25 were shared between the two shark species and the southern stingray (Fig. 2A). Forty-five orders were shared between all sample types (water, mucus and skin), 47 were shared between mucus and skin, and less than 20 were shared between tissue or mucus and water samples (Fig. 2B). Also, among elasmobranch species and types of samples, similar orders were found in every sample and with a similar distribution (Figs. 3A and 3B). The highest abundance was of the order Actinomycetales and the family Nocardiaceae (i.e.genus *Rhodococcus*), with a slightly greater abundance of reads obtained from the lemon shark and less abundance for reads obtained from the southern stingray. Mucus and skin samples had a higher number of reads than water samples (Fig. S2).

Phylogenetic diversity was used to build a rarefaction curve, following a phylogenetic aligned tree method, as it allowed a better visualization of data since most rarefaction

a. Species- Order

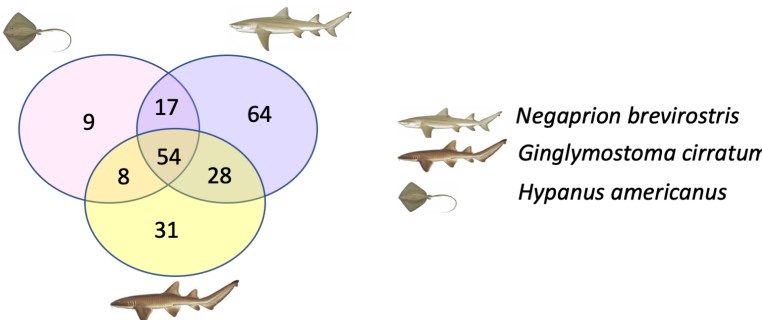

*Negaprion brevirostris*
*Ginglymostoma cirratum*
*Hypanus americanus*

b. Type of Sample- Order

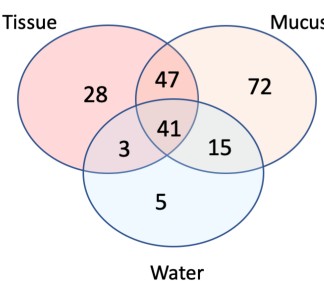

**Figure 2 Orders shared.** Orders (A) shared between and among elasmobranch species and those unique to each species. Orders (B) shared between and among sample types and those unique to each sample type.

curves were too similar to be distinguished. However, this methodology allowed clearer observation of the high heterogeneity found in samples used in this study, with some samples having a much higher number of total reads than others. Also, it showed that most if not all the samples did not reach an asymptote as showed in Fig. S3. For OTUs, Alpha-diversity was similar among species and among types of samples (Tables 1A and 1B). Alpha-diversity was non significantly different among species or among type of samples (Figs. 4A–4D).

Using the taxonomic category "order", the Bray–Curtis dissimilarity index, used as a $\beta$ diversity estimate, revealed greater dissimilarity (0.45) between the microbial communities found in the tissue and mucus of the southern stingray and those found in the shark species, the lemon shark, and nurse shark, and less dissimilarity (0.30) between the communities found in the skin and mucus of the lemon shark and nurse shark (Fig. 5). When each sample was used to calculate the Bray–Curtis dissimilarity index, patterns of bacterial

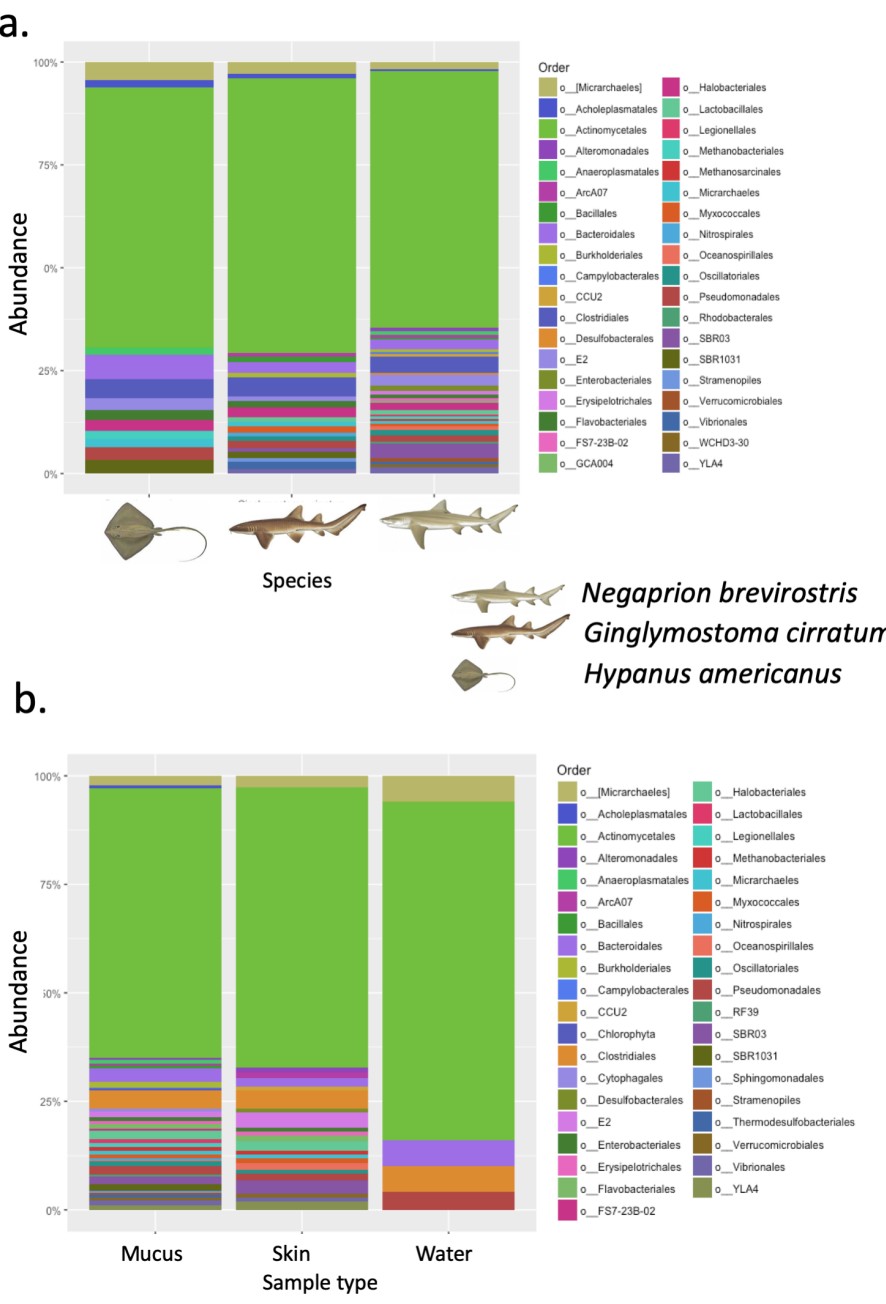

**Figure 3   Bacterial order composition.** Bacterial order (A) composition found for each elasmobranch species (*Hypanus americana*, *Ginglymostoma cirratum* and *Negaprion brevirostris*). Bacterial order (B) composition found for each sample type.

community dissimilarity were less clear, but it appears that mucus and skin samples from sharks and the southern stingray were less dissimilar from each other than when compared with the water samples (Fig. 6).

The percentage for each order identified from the total reads (sequences) obtained for each sample and analyzed is shown in Table S2 and Fig. S2. Few sequences were assigned to

**Table 1 Alpha-diversity indices for OTUs of bacteria found in this study (Shannon and Simpson) calculated by (a) elasmobranch species and (b) type of sample.** For Shannon and Simpson, the mean and range (in parenthesis) are reported.

| Species | OTUs | |
| --- | --- | --- |
| | **Shannon** | **Simpson** |
| (A) | | |
| *Hypanus americanus* ($n = 7$) | 8.06 (4.90–9.80) | 0.991 (0.966–0.998) |
| *Ginglymostoma cirratum* (Juveniles, $n = 7$) | 7.77 (4.70–11.72) | 0.984 (0.961–0.999) |
| *Ginglymostoma cirratum* (Adults, $n = 3$) | 9.08 (7.87–10.14) | 0.997 (0.995–0.999) |
| *Negaprion brevirostris* ($n = 15$) | 8.16 (5.19–12.12) | 0.993 (0.972–0.999) |

| Type of sample | OTUs | |
| --- | --- | --- |
| | **Shannon** | **Simpson** |
| Tissue ($n = 10$) | 7.41 (4.70–12.12) | 0.986 (0.961–0.999) |
| Mucus ($n = 18$) | 8.42 (4.90–11.72) | 0.993 (0.966–0.999) |
| Water ($n = 4$) | 8.66 (6.49–10) | 0.995 (0.997–0.999) |

the species or genus levels. Most of them were assigned to higher taxonomic levels (order). However, among the genus and species identified, several reported bacterial fish pathogens, symbionts and commensals were found in the mucus, tissue, and water samples (Tables S2, S4). It is interesting to note that some fish pathogens were only found in the water and not in the mucus/tissue samples, such as the order Pastereullales (i.e., *Pasteurella* spp., *Haemophilus* spp) and Oceanospirillales (i.e., *Halomonas* spp.)

The PCA showed higher similarity between the bacterial orders found in the skin and mucus of the two shark species in comparison with those in the southern stingray. Similarities were due to a similar number and distribution of reads identified as belonging to the order Actinomycetales (dimension 1) and to reads belonging to the order Bacteroidales (dimension 2) (Fig. 7). In the PCA separating each sample, including adults and juvenile nurse sharks, no clear differentiation patterns among microbial community compositions were detected (Fig. S4).

The co-occurrence analysis plot showed 104 out of all 202 recognized orders (including orders with less than 0.2% presence) (Fig. S5). Most of the correlations were between candidate orders, however orders such as Chlorobiales, Deinococcales Trembayales, Thermoproteales, Desulfarculales and Fusobacteriales showed strong co-occurrence. Orders such as Actinomycetales and Bacteroidales, highly influential in the principal coordinate analysis, where not found in the co-occurrence analyses plot.

# DISCUSSION

This study provides initial but useful baseline information on microbial communities in elasmobranch species from which changes in microbial communities over time and under changing conditions can be evaluated. Nevertheless, depending on the characteristics and populations of these animals, the composition and role of the whole community may vary. From a conservation perspective, knowledge of the microbial composition and function

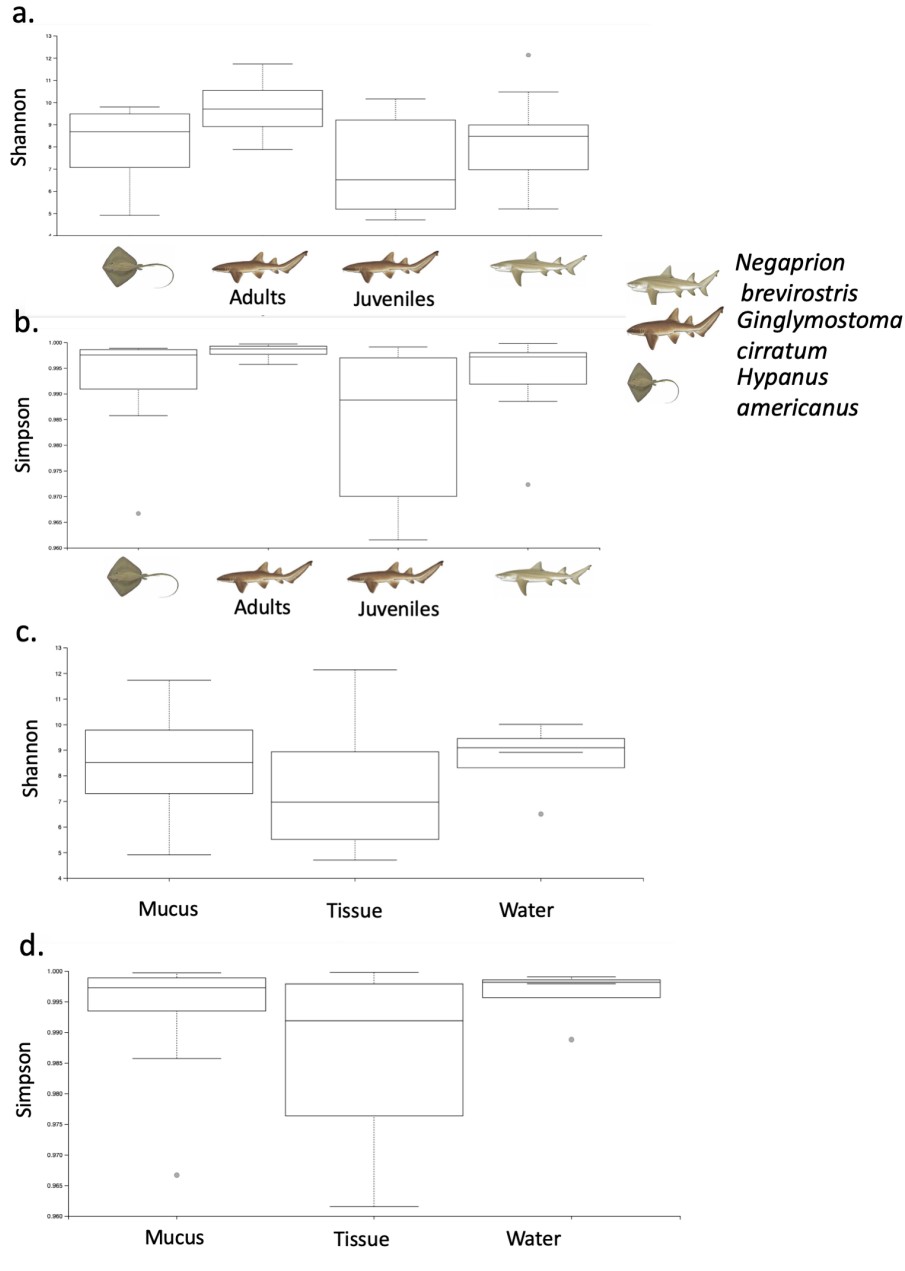

**Figure 4 Alpha diversity of OTUs.** Box-plots showing Alpha-diversity was non significantly different among species for OTUs (A) Shannon and (B) Simpson or among sample type for OTUs (C) Shannon and (D) Simpson.

may be an important approach for understanding how these organisms may be affected in the long term by environmental change; for example, climate change or ocean acidification (*Bahrndorff et al., 2016*).

In general, there were few taxonomically identified sequences compared to the total (only 16.9% of the total reads) and as compared to OTUs grouped (82% of the total reads).

Order

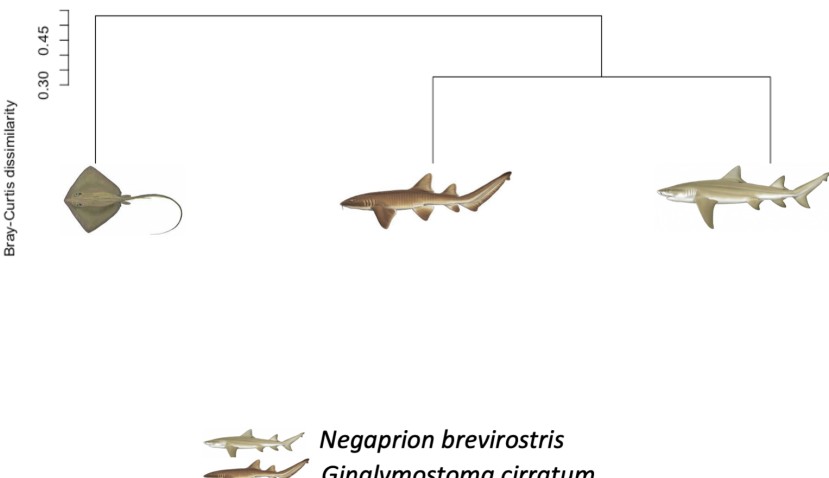

*Negaprion brevirostris*
*Ginglymostoma cirratum*
*Hypanus americanus*

**Figure 5** **Bray–Curtis dissimilarity index tree orders.** Bray–Curtis dissimilarity index tree showing greater dissimilarity (0.45) between the microbiome communities (for order) found in the tissue and mucus of the southern stingray and those found in the shark species, the lemon shark, and nurse shark, and less dissimilarity (0.30) between the communities found in the tissue and mucus of the lemon shark and nurse shark.

This may be the result of the shorter length of the sequences, the kits used in sequencing (to make libraries of short sequences), and differences in the DNA concentration at the start of the amplification and library preparation processes (*Solonenko et al., 2013*). Also, it has been suggested that the primers used in this study may amplify DNA from the host species (eukaryotic DNA), which would then reduce the total number of microbial reads that would have been included in our analysis (*Parada, Needham & Fuhrman, 2016*). However, our results are relevant to understand the microbial communities in elasmobranch fish and they suggest that skin tissue and mucus communities of the three elasmobranch species were similar in composition. Also, although some orders were shared with the water samples, more of them were shared between the two shark species and to a lesser extent with the southern stingray. Alpha-diversity at OTU level was similar among samples from the three species and among types of samples. However, there was high variation in the Alpha-diversity among samples within each species or within each sample type, which was confirmed by the rarefaction curve ran for all samples included in the study. This could be related with different number of reads obtained per sample, since alpha diversity indices can be sensitive to differences in sample sizes (*Barrantes & Sandoval, 2009*). This could have been caused by the storage conditions of some samples or due to the loss of DNA from some samples during the different steps of library preparation. Also, selective PCR amplification could generate higher amplification of some bacteria and not others. The richness of species was higher in mucus samples and in lemon shark samples. Composition

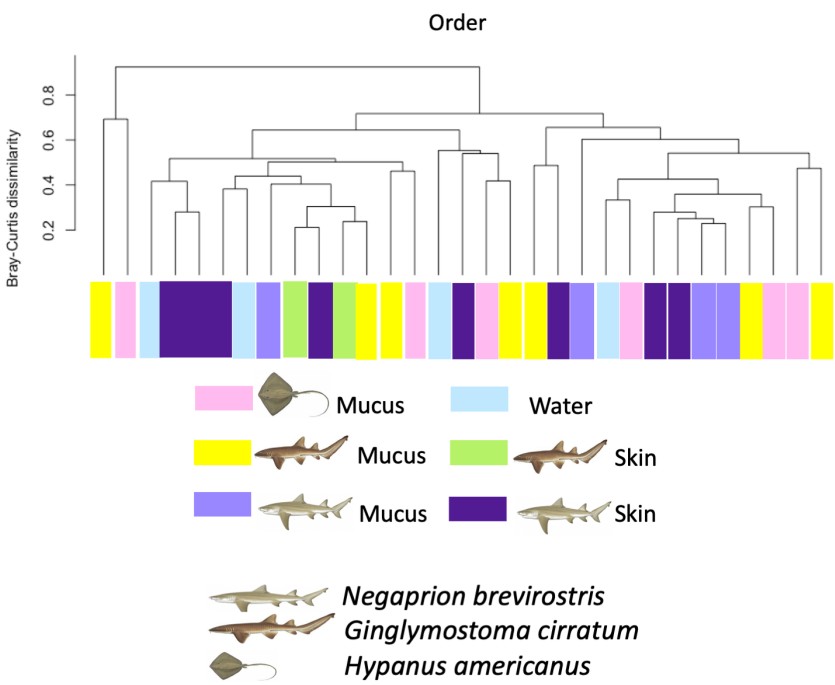

**Figure 6** **Bray–Curtis dissimilarity index calculated for order for different types of sample.** Bray–Curtis dissimilarity index calculated for order for different types of sample for different elasmobranch species showing mucus and tissue samples from sharks and the southern stingray being less dissimilar from each other than when compared with the water samples.

of mucus samples and skin samples from sharks tended to be more similar to each other than to the southern stingray or the water samples.

In this study, the bacterial diversity in the mucus and tissue included a wide range of orders, that have been described as pathogens, non-pathogens, and some that have scarcely been studied in relation to potential or confirmed hosts. Most orders identified belonged to the Bacteria Domain, with a very small proportion of Archaea. However, some of the Archaea identified in a mucus sample belonging to a nurse shark included Cenarchaeales, which have been found to be symbionts of one marine sponge that lives at very low temperatures (*Preston et al., 1996*). Interestingly, a high proportion of Actinomycetales (i.e., genus *Rhodococcus*) were found in mucus and tissue samples and influenced the community composition of all our samples, as showed in the PCA. Although Actinomycetales can be found in environmental samples from soil and water, and have been found in marine water and sediments, some strains have been isolated from marine environments and produce antimicrobial compounds against pathogenic bacterial and fungus (*Betancur et al., 2017*), particularly against some pathogenic strains of *E. coli* and *Pseudomonas* sp. (*Yellamanda et al., 2016*). The phylum Actinobacteria, to which the order Actinomycetales belongs, has also been found in the skin microbiota of bony fish (Osteichthyes) (*Larsen et al., 2013*). The finding of some Actinomycetales also in the water samples may represent contamination, although as mentioned before, they are an abundant order in marine environments. Also, a negative control was always included in the PCR amplifications and came out clean in

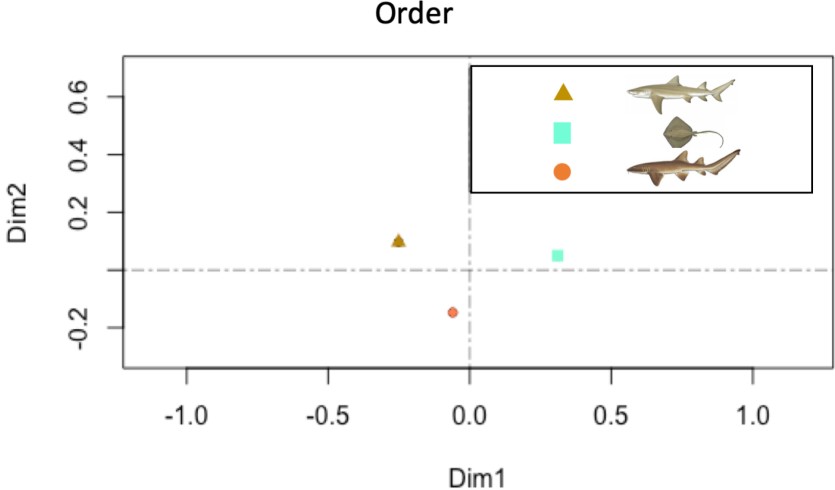

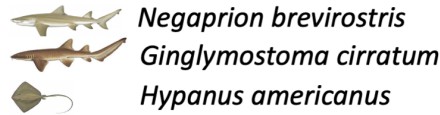

*Negaprion brevirostris*
*Ginglymostoma cirratum*
*Hypanus americanus*

**Figure 7** **Principal component analysis (PCA).** Principal component analysis (PCA) showed higher similarity between the bacterial communities found in the tissue and mucus of the two shark species in comparison with those in the southern stingray at both order and family levels. Similarities were due to a similar number and distribution of reads identified as belonging to the order Actinomycetales and to the family Nocardiaceae (dimension 1) and to reads belonging to the order Bacteroidales (dimension 2).

all cases. Also, by not including OTUs or orders with less than 0.2% representation in the total sample, we tried to control for possible contaminants in the samples, maybe due to manipulation in the field or in the laboratory setting.

Although we focused our analysis on microbial community diversity and composition of the orders identified, we also investigated their characteristics and those of the genera within each order because, although a smaller number of reads were identified to the genus level, some interesting data was obtained. Within the bacterial order and genera found only in water samples, three have been described as pathogens for fish, including the order Pasteurellales (genus *Pasteurella* spp. and *Haemophilus* spp.) and of the order Oceanospirillales (genus *Halomonas* spp.) (*Bullock, 1961*; *Hawke et al., 1987*; *Austin, 2005*). There was also a species only found in water samples, *Acinetobacter johnsonii* (order Pseudomonadales), which has been described as a fish pathogen (*Kozińska et al., 2014*). Other sequenced bacteria present in the results of water samples, such as *Moraxella sp.*, are opportunistic bacteria and have been found in other animals, for example in mammals (*Whitman, 2015*). Some orders found only in the elasmobranch samples may also play a role as pathogens. The order Alteromonadales (i.e., genera *Alteromonas, Shewanella*) (*Boone & Bryant, 1980*), Actinomycetales *(i.e., genera Mycobacterium* and *Nocardia),* Bacillales (i.e., *Staphylococcus*) and Flavobacteriales (i.e., *Chryseobacterium*) have been reported

as pathogens for various fish species (*Hansen & Olafsen, 1999*; *Austin, 2005*). The order Syntrophobacterales (i.e., genus *Syntrophobacter*) was also present in mucus and skin samples and considered a possible pathogen for fish, due to the fact that bacteria that belong to this group, degrade propionate, a corticoid used in healing skin (*Schulze et al., 2006*). However, many other Flavobacteriales (i.e., *Flavobacterium*), Vibrionales (i.e., *Pseudoalteromonas*), Lactobacillales (i.e., *Lactobacillus*) and Bacillales (i.e., *Bacillus*), also found only in elasmobranch samples, are considered symbionts of marine fish (*Anand et al., 2011*; *Luer et al., 2014*). Some species of *Flavobacterium* have been studied as commensal to fish, and have shown antimicrobial activity against fish pathogens from the genus *Vibrio* (*Lal & Tabacchioni, 2009*). *Bacillus polymyxa*, found in mucus and skin samples in this study, has been isolated from fish guts and some strains of this species synthesize antibiotics (*Olmos, 2014*). Similarly, *Bacillus subtilis* has been suggested as a probiotic involved in the optimization of fish feeding (*Merrifield & Rodiles, 2015*). Finally, various orders sequenced from mucus and skin samples are considered normal biota of fish gills or skin (i.e., Xanthomonadales, Caulobacteriales) (*Sugita et al., 1996*).

However, it is important to remember that pathogenicity may be related to particular strains (*Fitzgerald & Musser, 2001*) so caution is needed in the interpretation of these results. For example, three orders found in mucus and tissue samples Lactobacillales (i.e., *Streptococcus* and *Enterococcus*), Pseudomonadales (i.e., *Pseudomonas*) and Vibrionales (i.e., *Vibrio*) are sometimes reported as pathogens and sometimes reported as symbionts. For example, *S. parauberis* produces streptococcosis in some fish (*Austin, 2005*; *Nho et al., 2009*; *Jung & Rautenschlein, 2014*; *Abrahamian & Goldstein, 2011*), but other *Streptococcus* spp. inhibit the growth of pathogenic bacteria (*Hansen & Olafsen, 1999*). Similarly, *Pseudomonas putrefaciens* acts as a pathogen for fish (*Abrahamian & Goldstein, 2011*), but *P. fluorescens* inhibits growth of pathogens (*Subramanian, Ross & MacKinnon, 2008*) and has been isolated from healthy salmon eggs and mucus (*Cipriano & Dove, 2011*; *Akinyemi et al., 2016*). Finally, *Vibrio* have been reported several times as an important pathogen for marine life because of its great capacity for survival and of acclimation in its host, as it hydrolyzes urea and uses it as a source of carbon and nitrogen (*Hansen & Olafsen, 1999*). Many species have been described as infectious for *Negaprion brevirostris*, especially when they are physically injured (*Grimes et al., 1984a*; *Grimes, Gruber & May, 1985*); others are associated to the mortality of sharks in captivity (*Grimes et al., 1984b*), and others to infections caused by hooks (*Borucinska et al., 2002*). There are some species that, depending on the strain, are pathogenic or not, such as *V. alginolyticus* and *V. parahemoliticus* (*Austin & Austin, 2007*; *Abrahamian & Goldstein, 2011*). Other species, such as *Vibrio alginolyticus* and *V. fluviales,* are considered pathogenic for fish in general (*Zorrilla et al., 2003*); *Vibrio fortis* has been reported as a sea horse pathogen (*Wang et al., 2016*); *Vibrio shilonii* has been found to cause coral bleaching (*Kushmaro et al., 2001*).

There are various bacteria identified in the mucus samples that are considered in other studies as symbionts or pathogens for other animals or humans. For example, some species of the order Bacteroidales (i.e., *Bacteroides*) have been described as human pathogens in periodontal disease and *Prevotella copri*, found in mucus and skin samples have been identified as pathogens in intestinal inflammation. Additionally, bacteria from the order

Clostridiales (i.e., *Helcoccocus*) have also been described as pathogens for humans (*Chow & Clarridge, 2014*). Also, many species within the order Chlamydiales are reported as pathogens for birds and mammals (*Whitman, 2015*).

As examples of symbiosis of species of bacteria (found in samples for this study) with humans or other animals, it is worth mentioning *Lactobacillus zeae* (order Lactobacillales), which has been found to serve as protective biota for nematodes (*Zhou et al., 2014*); *Butyrivibrio* and *Selenomonas* (both from the order Clostridiales) are found in the gastrointestinal tract of ruminants; other members of the order Clostridiales, including *Faecalibacterium prausnitzii, Peptoniphilus, Ruminococcus, Megamonas* (*Chow & Clarridge, 2014*) and *Butyricimonas* (from the order Bacteroidales) (*Wexler, 2007*) are normal important bacteria in the human gut microbiota. Other orders sequenced from mucus samples were Flavobacteriales such as *Sulcia muelleri* (*Moran, Tran & Gerardo, 2005*), Enterobacteriales such as *Baumannia cicadellinicola* (*Cottret et al., 2010*) and Trembayales such as *Carsonella ruddii* (*Thao et al., 2000*), which have been described in symbiotic association with insects. A very interesting case is the order Burkholderiales (i.e., *Janthinobacterium lividum),* which has been found in the skin of some amphibians and appears to prevent infection by *Batrachochytrium dendrobatidis* (*Brucker et al., 2008*). These are startling examples that may be related to the findings of this study; however, more in-depth research should be conducted to identify the pathogenicity or symbiosis properties specifically in elasmobranch or fish.

Results from the co-occurrence analysis presented some interesting results but not clear patterns related to the PCA results or to other previously presented analyses. Strong co-occurences were found among orders such as Elusimicrobiales, Halanaerobiales, Synachococcales, Solibacterales which are defined as marine environmental bacteria (some of these orders can be classified as cyanobacteria), including some desulfurating bacteria, such as Desulfarculales, but also with bacterial order characterized by their presence in extreme habitats, such as Thermobaculales and Thermoproteales. This could suggest that either these are random co-occurrences among environmental bacteria that may be contaminants to the mucus and skin samples or that desulfuration may be an important metabolic path used by bacteria in these microbial communities as has been shown in the gut microbial communities of some marine fish (*Egerton et al., 2018*). Further research on this idea may be warranted. Interestingly, Fusobacteriales, a bacterial order which has been previously found in the human gut (*Suau et al., 2001*) as well as in the gills of coral reef fish (*Reverter et al., 2017*), as well as Trembayales, an order of bacteria found to be endosymbionts of insects (*Thao et al., 2000*), were also found in the co-occurrence analyses, suggesting a possible role in the skin and mucus microbial communities of elasmobranch.

According to this study, the role of the mucus and the bacteria associated to it may depend on numerous variables, including the virulence and pathogenicity of each microorganism (*Hansen & Olafsen, 1999*). Opportunistic bacteria can acquire virulence determinants with environmental changes by different means, for example, by (a) increasing their numbers by exploiting the higher production of mucus (glycoproteins) induced by presence of toxic substances in the water (*Hansen & Olafsen, 1999*), by (b) shifting from a non-infectious state to an infectious one through an activation caused by a physical or chemical change in

the environment (*Hansen & Olafsen, 1999*), or by (c) Reaching the dermal layer to infect the host taking advantage of a degree of reduction of the defensive mucus layer, caused by the presence of abrasive substances in the surroundings of the fish (*Benhamed et al., 2014*). These three opportunities for the bacteria to infect the hosts not only benefit these microorganisms but they also affect the host by reducing their physiological condition (*Austin, 2005*), and may explain the finding of the reported bacterial pathogens on the skin of healthy animals.

The orders considered fish pathogens found in the water samples but absent in the elasmobranch samples, allows this research to present an interesting assumption. We suggest that there may be specific antimicrobial activity in the skin environment, or partial control against infections that exists in low concentration in the mucus, but this might be also a result of the low number of samples and replicates analyzed (*Rakers et al., 2010*). However, it is very likely that difficulties in sampling -for example, handling the sharks and stingrays-, prevented us from collecting a larger skin or mucus sample and that this in itself could be biasing our results.

The simultaneous presence of pathogens and possible symbionts varied between samples; however, the role of each order should be verified for each of the host species considered in this analysis. According to these results, we suggest that the role of the epithelial microbiota may be considered as a first line of defense against infectious organisms but it could also be a potential threat for the injured host. This may be particularly relevant as a protective mechanism for sharks and rays that get hurt during copulation and that could otherwise die due to infected wounds. This could also depend on the whole combination of bacteria and their interaction between them in each host, as well as with the host cell and physiology, known as the "holobiont" (*Carthey et al., 2020*). As mentioned earlier, each fish may accumulate a specific community of microorganisms in its life span depending on the environments it inhabits during its development and growth (*Hansen & Olafsen, 1999*). This particular accumulation and interaction between the microbiota and the host (holobiont) may also affect aspects such as survival and reproduction of the host, and may become relevant for conservation of these shark species in the near future (*Carthey et al., 2020*).

This study represents the first contribution to describing shark and ray skin and mucus microbial communities. The next steps to further understand the role of bacterial communities in skin and mucus of elasmobranchs require functional metagenomics and metabolomics analyses to unveil the role of these bacteria.

## CONCLUSIONS

This study presents the first description of skin and mucus microbiota from two shark species and a stingray. Orders were highly diverse and similar between species and types of samples and a higher number of orders were found in skin and mucus when compared to water samples. The order Actinomycetales was found in a very high percentage (>50%) of skin and mucus samples and could represent bacteria that may have antimicrobial activity, however the co-occurrence analysis showed strong relationships among order previously

found in the human and fish gut, as endosymbionts of insects and among orders involved in metabolic paths related to desulfuration. This is baseline information that could help in future monitoring of microbiota change in elasmobranch species that may be caused by climate change and ocean acidification.

## ACKNOWLEDGEMENTS

We would like to give a special thanks to all those involved in samples collection, particularly to the Bimini Shark Laboratory, to D Cardeñosa, and to R Vieira and his team at Oceanario Islas del Rosario (Ceiner). We thank E Salguero and AP Jimenez for their help with sequencing.

### Funding

Financial support for this project was provided by Proyecto de Ciencias Básicas, Vicerrectoria de Investigaciones, Universidad de los Andes. The funders had no role in study design, data collection and analysis, decision to publish, or preparation of the manuscript.

### Grant Disclosures

The following grant information was disclosed by the authors:
Proyecto de Ciencias Básicas, Vicerrectoria de Investigaciones, Universidad de los Andes.

### Competing Interests

None of the authors of the manuscript have identified any professional or financial competing interests. None of the authors is employed by a governmental institution. Martha J. Vives is an Academic Editor for PeerJ.

### Author Contributions

- Susana Caballero conceived and designed the experiments, analyzed the data, prepared figures and/or tables, authored or reviewed drafts of the paper, supporting material and all figures. Computational work, and approved the final draft.
- Ana Maria Galeano conceived and designed the experiments, performed the experiments, analyzed the data, prepared figures and/or tables, authored or reviewed drafts of the paper, computational work, and approved the final draft.
- Juan Diego Lozano analyzed the data, prepared figures and/or tables, authored or reviewed drafts of the paper, computational work, and approved the final draft.
- Martha Vives conceived and designed the experiments, prepared figures and/or tables, authored or reviewed drafts of the paper, suggested reviewers editorial revision, and approved the final draft.

### Animal Ethics

The following information was supplied relating to ethical approvals (i.e., approving body and any reference numbers):

The Animal Care Committee at Universidad de los Andes (CICUAL) approved this study.

## Data Availability

The 16S datasets used in this article with accompanying metadata are available at Dryad: Caballero, Susana; Galeano, Ana Maria; Lozano, Juan Diego; Vives, Martha (2019), Description of the microbiota in epidermal mucus and skin of sharks and rays, Dryad, Dataset, https://doi.org/10.5061/dryad.b5mkkwh8j.

The data is also available at Figshare: Caballero, Susana; Galeano, Ana Maria; Lozano, Juan Diego; Vives, Martha (2020): Description of the microbiota in epidermal mucus and skin of sharks (Ginglymostoma cirratum and Negaprion brevirostris) and one stingray (Hypanus americanus). figshare. Dataset. https://figshare.com/articles/dataset/Description_of_the_microbiota_in_epidermal_mucus_and_skin_of_sharks_Ginglymostoma_cirratum_and_Negaprion_brevirostris_and_one_stingray_Hypanus_americanus_/6863765.

Sequences are available at GenBank: SAMN16838150 to SAMN16838180 in BioProject PRJNA679570.

## Supplemental Information

Supplemental information for this article can be found online at http://dx.doi.org/10.7717/peerj.10240#supplemental-information.

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
