# Peer review of "Description of the microbiota in epidermal mucus and skin of sharks (Ginglymostoma cirratum and Negaprion brevirostris) and one stingray (Hypanus americanus)"

_PeerJ, doi:10.7717/peerj.10240_

## Round 0.1 · original submission · Major Revisions

Please, carefully address the reviewers comments, in particular regarding clarity of writing, and details for the bioinformatic methods.

·

Basic reporting

The focus of the manuscript is very interesting, the role and composition of microorganisms in the skin mucosa of fish is rarely studied. However, the analyses used and the results and discussion need to be improved. The main aims and objectives could be made clearer and the advancement made through this research could be highlighted further. I liked Figure 1 - would be nice as a graphical abstract. The figures/tables shown could be improved.

In general there are very few grammatical errors. But watch out for double spaces after full stops. Check that the line spacing is the same between each sentence.

Abstract

Line 2: Remove the word “as”.
Line 11: 16S rRNA gene.
Line 11-12: Next generation sequencing does not target the whole 16S rRNA gene. It is amplicon sequencing? What region of 16S was targeted using your specific bacterial primers? Was it a universal primer set? What sequencing technology? Pyrosequencing? Illumina Miseq?

The background sentences could be shortened. The abstract needs to be more conclusive and highlight the merit of the study.

Introduction

Line 44 – 47. The reference on line 44 can be removed as you can leave it at the end.
Line 50: Appears.
Line 55: Definition of polyandry?

Experimental design

Methods

Could be a bit clearer on how skin tissue samples were taken. Were the samples weighed? Was the same size/depth taken? These are all factors that will influence your DNA extraction and thus your sequencing results. How did you extract from the water samples? What method used?

Line 96: These primers are a universal primer set which amplify bacteria and archaea.

What error rate in the sequencing? How many sequences removed after quality control?

Line 101: Remove “The” and one “was”.
Line 104: Why were only 32 used? How did the DNA concentration determine what samples? Did some not amplify? Be more precise in your methods section.
Line 107: Quantified.

Why pairwise dissimilarity performed? Why Horn? Would Bray-Curtis dissimilarity matrix not have been more appropriate? Also consider stats that include Unifrac and Weighted Unifrac to consider phylogenetic distances. Was the data rarified or transformed in any way? How many total sequences were found?

N.B. You need to submit the raw sequences to Genbank and give the accession number.

Validity of the findings

Results

Line 139: Were they Archaea? Your primers target both.

Your first paragraph reads more like a methods section. One sentence should suffice, total number of sequences how many Phyla/OTU identified etc.

The results and discussion section need a lot of improvement. You need to consider species richness/evenness/ - alpha-diversity/ beta-diversity. Perhaps look to other microbial ecology manuscripts to give ideas to strengthen the results and discussion sections. Figure 2 should be in terms of the relative abundance in % terms so you can compare across the samples. Focusing on phyla level does not really give a lot of information. Table 1 could be supplementary. Is , meaning a decimal place? 3,182 reads from 32 samples is very very low. Consider investigating the sequence analysis further. How many total reads per sample before and after QC? Error rate of platform? More in depth analysis needed.

Additional comments

In general I think this is an interesting manuscript but the methods section needs a lot more detail. I think your sequencing analyses should be repeated and more rigorous analyses performed. The presentation of the data could be improved.

·

Basic reporting

-While I did not factor this into my review, the paper needs a thorough editing. It is difficult to follow the author’s train of thought at times.

-The methods, in particular the bioinformatics, are very unclear and need revision for clarity.

-Not a single statistic is used in this manuscript. It is hard to draw conclusions without some significance testing.

-Overall, the figures need polishing they look straight out of R.

-Did the authors normalize their data for analyses? Given the very wide distribution of sequencing depth (fig 2), this should be done for beta diversity analyses.

-My main concern with this study is the data and the massive loss of data the authors report. The authors report 219k raw reads (line 136) and following filtering they use ~3.6k. What happened to the rest of the data? Was this just a bad run? What did the authors do to filter the data? More information is needed to evaluate this data.

Experimental design

This is fine overall. A better description of the methods is needed however

Validity of the findings

The results are reasonable. But it is hard

Additional comments

Line 80: please provide GPS coordinates for the sampled area. Also, a table would be nice summarizing the species and # samples.

Line 83: please provide GPS coordinates here too.

Line 87-88: how was the water collected and filtered.

Line 91: how long were the samples kept at 4C?

Line 95: I would suggest combine these two sentences into one

Line 97: 515/806 only amplifies the V4 region of 16S. Also I suggest moving ‘region’ to after ‘v3-v4’ to make the sentence flow better.

Line 97: capitalize the ‘s’ in 16S

Line 99: was this a single PCR reaction for each sample?

Line 101: this sentence needs to be rewritten. Also, outside of the water could the authors actually view skin metaG DNA on a gel?

Line 111: where was the sequencing performed?

Line 114: what parameters were used to filter the data?

Line 117: <20 sequences per sample doesn’t indicate poor quality, just poor sequencing depth. Do the authors mean a PHRED score <20?

Line 119: please elaborate on the methods used in QIIME

Line 120: OTU does not mean taxonomy. Do the authors mean the clustered the reads in OTUs and the assigned taxonomy? What methods did they use?

Line 131: did the authors deposit the sequences in a database (e.g. NCBI).

Line 120: what version of QIIME?

Line 122: ggplot is only a plotting package, not for analyzing diversity. Did the authors calculate diversity?

Line 124: I’m not sure what the authors are trying to say in this paragraph. They say MDS and then Horn similarity/PCoA. What analysis was used, make this clear. Also was the data normalized?

Line 129-130: this likely isn’t needed.

Line 140: should be ‘Bacterial’ not ‘Bacteria’.

Line 156: species-level assignments are hard with this dataset given the small fragment length (~250bp). I would suggest the authors avoid this type of analysis.

Line 161: did the authors try to add statistics to back this paragraph up?

It would be nice to talk about fish species difference in results/methods.

Discussion: the authors should, if they are going to talk about genera, provide some data showing the genera and not rely on the text to inform the reader.

Line 287: the authors do not report diversity estimates anywhere in this paper.

Figures and tables
Figure 1: In the caption, OTU’s should be OTUs. When the authors say OTUs at the phylum level, do them mean shared phyla? If so then they should revise the legend accordingly.

Figure 2: the color scheme on this is very taxing on the eyes, the colors are too similar to
discern between the phyla present. I would suggest using a different palette and simplifying it
to smaller number of taxa.

What does the y-axis mean? Number of reads?

The depth of sequencing is clearly different between samples so it makes comparing between
samples impossible. I would suggest the authors that the output from the QIIME script
“summarize_taxa.py” and use the relative abundances to make this plot.


Also, the phylum level is very coarse. It tells the reader very little about the system given the
diversity of functions bacteria are capable of in a single phylum. I would suggest the authors
do this at the order or family level to provide more information about the data.

What do the boxes around the bars mean?


Figure 3: is this a PCA or PCoA? The axes say PCoA, the legend says PCA and the methods say PCA. What method was used?

---

## Round 0.2 · Major Revisions

The submission has been re-reviewed by 2 of the previous reviewers. Please address their concerns in a revision\nOne of the reviewers asked me to ask you to deposit your sequences in a public database and add accession numbers to your manuscript\n\nMy own comments : Since the one of the first round reviewers did not accept to review the resubmission, I see you answer his/her main concern by adding the number of sequences passing the filtering steps, but I would like you to comment WHY such a very small number of sequences are left at the end (i.e. was that a particularly bad run?). Also I am not super familiar with Ion Torrent data treatment (I did work with 454 data and we used to denoise it), or the mothur pipeline, but the details of your data analysis are not well explained and in my view have some significant problems. For example, the paper you used as reference to the SILVA database is incorrect, so I am not sure which version of Silva you used for the analysis and more important you do not explain the criteria to which you use to keep and eliminate almost 20K (~85\% to good reads) sequences (i.e. how did you evaluate « poorly aligned » or « unassigned », so there is no way for someone else to reproduce your results. This point is actually critical since it can change you interpretation of the results).\n\nI ask you to please address these concerns in your resubmission

·

Basic reporting

Thanks to the authors for making the suggested changes. I appreciate the effort that went into the reanalysing of the sequencing data and the manuscript has improved considerably. I have revised my suggestion however to still consist of major corrections based on these changes to further improve the quality of the manuscript – but these should not be too difficult to address but would represent more than minor changes.

Your raw data needs to be submitted and given an accession number, it is possible to not make this public until the manuscript is published but it needs to be submitted before accepting for publication. Perhaps the editor can advise if you need to submit a letter if you want to avoid submission to a database for a specific reason?

1) Can you include OTU-level analyses?

"were assigned taxonomically to OTU against the SilvaV4 database"

OTUs are classified based on a cut off threshold -usually 97% similarity with each other. Is this what you did? The sequences of these OTUs are then aligned to the specific database you choose which is a separate step. Thus, OTUs are independent of the database and a true representation of your samples. Considering many OTUs could not be matched to the database including OTU-level analysis could provide a detailed picture of the microbiology of your samples.
-->The alpha and beta diversity measures in Table 2a and 2b don’t provide a lot of information. The authors discuss within the text but fail to address whether there are significant differences between the groups or samples? From Figure 4 it appears there is no significant difference? If so can add to Supplementary Information. Is there a difference between the water and the three fish types sampled? Also was the data rarefied to the lowest sequencing depth sample? These would make more sense performed at OTU-level ie. Not taxonomically assigned to the database.
-->Rarefaction curves need to be for each individual sample and ideally at OTU level rather than taxonomic assignment to order or family level.
-->Discuss diversity before discussing what is shared and any description of taxa found.

2) Can you describe your sequencing QC measures, each step and the sequences left after each output? Were chimeric sequences removed?

3) Figures

-->8 Figures is a lot to include. Are all these essential? Are all these necessary to back up your conclusions in the manuscript? -Table 1 is too large for inclusion within the manuscript. This could go in Supplementary Information along with the number of sequences that passed QC for each sample type. Can include a taxa plot (species above 1-2%) at family level for each individual sample instead of table?
-->For the shared Venns display and discuss one level only – or else OTU and then family (instead of order and family as a lot of your OTUs could not be assigned). Describing both family and order is confusing in these cases and adds no extra value. In methods you say OTUs but you show only show taxa.
-->Can PCA include all samples? PCA (Principle Component Analysis) and PCoA (Principal Coordinates Analysis) measures are different, which was used?

Experimental design

No comment

Validity of the findings

No comment

Additional comments

Abstract:
Line 11: Remove sentence about collecting water samples.
Line 18: change to ‘and there was no significant difference’.
Line 21: Add the water collection sentence to here. Combine as one shorter sentence. E.g. sampling of the raw water revealed…..These were not observed in.
Line 21-23: What is the evidence for their function?

“We found potentially pathogenic bacteria in water samples such as Pasteurella spp., Haemophilus spp. and Halomonas spp. but these were not found in the tissue or mucus samples from any species. We found some bacterial groups such as Flavobacterium, Pseudoalteromonas, Lactobacillus and Bacillus that could play a role protecting the animals from pathogenic infection”. -Can you back this up with further data? Perhaps a co-occurrence network analysis?

Results:
Why would you expect the water to have a lower diversity? Any evidence? Less DNA quantified?

Discussion:
Line 326: Remove sentence re. startling examples.
The discussion largely discusses genera identified but the results you display are not at genus level.

-->Some of the colours in your taxa plot colour scheme are too similar. The text for these colour codes needs to be increased in size.
-->Figure 6 and 8 say Order and Family? Which one?
-->Include the legend for the fish pictures in each figure where these are used.

Reviewer 3 ·

Basic reporting

The English used in the manuscript is correct, but sometimes there are long sentences that impede to understand everything at the first reading. I suggest to authors to make a review by a native speaker.
The references used in the article are very appropiated, and the structure of the article is correct for the publication.

Experimental design

The methodology used for perform the study is appropiated.

Validity of the findings

It is the first study assessing the microbiota of mucus and skin in 3 Elasmobranchs species, which is interesting to have a reference in future studies.

Additional comments

In the manuscript entitled “Description of the microbiota in epidermal mucus and skin of sharks and rays”, authors performed a description of the microbial communities from 3 Elasmobranch species. It is very interesting to have a reference for future studies, and thus this manuscript deserve to be published. However, it still needs some more work to clarify some aspects of the study.
One of the main concern that I have is about the different samples belonging to the nurse sharks. There are two blocks of samples: wild juveniles and captive adults. Both factors (habitat and age) may influence the microbiome, so I think that it is not correct to include those two groups as the same kind of samples just because they belong to the same species. Actually, if it is the environment which shape microbial communities, we could expect more similarities between the two shark species sharing the environment in the sea, than among wild juveniles and captive adults of the same species. Those factors should be taken in consideration in the analyses, because I’m sure that they are going to explain a part of the variance in the nurse sharks. However, it won’t be possible to distinguish among both factors since there are just those 2 categories.
I have major concerns with the statistical methods. Authors say that they performed Kruskal-Wallis tests, but they didn’t present results, in the general comparison nor in the group by group one. I guess that they decided if results were significant by looking at the box-plots, which is absolutely wrong. They speak about differences in diversity of different categories, but they don’t support this with statistics. In the Fig. 4 they say that there are no visual differences, but I don’t agree, since for example in E we could say that there is. But everything is speculation, if they don’t show the statistical results.
Another minor question is that when presenting the study, authors say that they worked on the taxonomical categories of Order and Family. However, in the Figure 1 for example, but also in the Discussion, they include the Phyllum and / or the Genus. Everything should be consistent. If you decide to work just on Order and Family, then remove results apart. And if you decide to include them, then say from the beginning.

More specific comments:
L6: change to “from the water or from males’ mouth”
L6: “The role…” this sentence should be placed at the beginning, before talking about sharks, and as a general feature of fish. “…from colonizing the skin. However, the role…”
L9: “Tissues” is confounding. Authors has analysed just the skin, so I suggest to change tissue for skin all along the manuscript.
L14: “we analysed sequences… species” this sentences could be removed.
L16: “particularly” is in some group, not in all the fish samples. Maybe say the diversity was higher in fish than water. And remove “was high” if you don’t say in comparison to what? There are some reference values of diversity?
L18: comparatively to what?
L26: “changes in… conditions, including…”
L32: I suggest, for more clarity, to start the introduction as follows: “The first barrier of protection against microorganisms in fish is the mucosal immune system”
L71: remove “in the process”
L76: “…mucus, and the skin tissue, to assess differences among the fish microbiome and this of the surrounding water.”
L77: Change “Doing so…” to “This will help to understand the role…”
L78: “…we characterized the bacterial…”
L79: change from to in
L87: how can you know that they are healthy individuals? In those fish captured in the ocean, it is not possible to know. Maybe you could say "apparently healthy", or explain why you know they were healthy instead.
L95: More details on the sampling should be given. For instance, how did you cut the skin? With sterile scissors or could have had contamination among samples? How long lasted the manipulation? Fish were immediately released?
L100: continue breathing
L104: “and PCR amplification” in italics.
L105: “from all the entire samples collected” Remove in L106 “In all cases…”
L106: remove “for all samples”
L107: at which diameter of filtration? What does it means concentrated?
L108: include the sequences of the primers, and a reference.
L110: temperatures should be written as the first one: 94 ºC, 50 ºC.
L117: insert space 1.5 %
L133: (skin, mucus or water)
L140: comma after “family”
L146-152. It’s messy. You speak of normality, then Venn diagrams, and then Kruskal-Wallis. The normality analyses should be together with the Kruskal-Wallis since is part of the same.
The Venn diagram is not a statistical analyses, is just a representation of descriptive data, so I don’t think that it is necessary to say here.
Also, it is not clear if the K-W test was applied to all or only to the data that did not fit in a normal distribution.
Finally, it is not necessary to say that data were visualized in box-plots since this is part of the K-W test.
L179: It’s messy, the diversities are mixed. Say first the differences among fish and sample type, making the comparison of diversity among different samples, and then bacterial groups.
L193: which is the reason to expect a higher diversity in the fish than in the water? I think that is plausible to expect the opposite.
L210: change “appeared to be” to “are”
L221: The first speculative sentence should not be the beginning of the Discussion. It should be removed, or placed below, and start with “this is, to our knowledge…”
L239: I guess is the average of the alpha-diversity, if the variation intra category is not significantly lower than inter categories (statistics?)
L252: Cenarchaeales, which…
L258: change which to and
L264: analysis on microbiome
L267: were obtained
L279 and so on: In general, there is always a controversy in deciding if a bacterial taxon is pathogenic or not, because sometimes it depends not just in the species, but in the strain. Here for example, Bacillus include numerous keratinolytic groups, which may cause wounds in fish skin.
L288: parenthesis
L371: “presents the first description of skin and mucus...”
L642-643: It’s OTUs, no OTU’s
Fig. 2. In the text, the figures are cited in minuscule (2a, 2b…), but in the graphs they are in majuscule. Be consistent along the manuscript.
In the name of the graphs, please include Fish species instead of species, because it could be confounded with bacterial species.
Fig. 4. Statistical results are needed.
Fig. 6. I don’t understand the legend and figure. It supposed to be the distance among different categories but I can’t see this in this graph, not differences among sample types. It’s like if they are mixed, is it the average?
Fig. 7. The skin of the stingray is absent?

---

## Round 0.3 · Minor Revisions

I am sorry for the delay on the decision, due to issues linked to confinement in France. As you can see both reviewers appreciate the improvement in the manuscript but asked you to increase the taxonomy resolution of some of your analysis. I would ask you to address all their comments in a revised version

·

Basic reporting

Third Revision: Comments in attachment

Experimental design

No comment for this version

Validity of the findings

Line 499: Bioinformatics method – in your results you have the numbers of reads and how many remained after QC. Please add to this section. A lot of reads didn’t pass the QC. Mothur should tell you how many samples remain after each QC step and chimera removal please include if possible. Comment in the discussion about the high error rate and relate to the literature on the platform and other studies. I believe these primers (Caporaso) when used for salmon pick up a lot of host DNA and so have a high error rate so could you comment on the source of the read loss (ie chimeras, read length, host DNA contamination etc).
Recommended vevision: Sentences in discussion discussing error rate where you already discuss low % of assigned species. More detail on QC.

I don’t agree with having diversity measures to order level I think it is too high. In the previous version there was inconsistency between figures and text showing different levels so understandably choosing one level is easiest and is clearer, but for these analysis the standard is OTU-level or genus maybe family-level. For example in the results it is stated ‘Alpha-diversity was similar among species and among types of samples’ I’m highlighting this as I think that is likely you are just not looking deep enough and missing something interesting in your study. Unfortunately many of your reads did not assign at genus/family level. As you state only 16% of your reads were assigned taxanomically.
Recommended vevision: Test alpha-diversity at OTU level.

The rarefaction results I would like to see at read-level to see the variation per group but also per sample. Especially given how many reads were removed through QC. From reading the manuscipt there’s no way of knowing if this is even per sample. The rarefaction at this level is crucial here. In fact you mention ‘samples having 10 times the number of different orders found in other samples from the same type. Something similar was found for the samples from each species, meaning that there was high heterogeneity in samples used in this study (rarefaction curves for each species and sample type’ Supplementary Figure 2)
Recommended vevision: Rarefaction curve with reads after QC or OTU level per sample.

For the rest of the analysis it is fine to be at order level. The description of interesting genera in the discussion reads well.

Additional comments

Getting a paper to publication can be challenging at times and I’d like to thank the authors for making significant changes over a considerable time-frame. Overall these have improved the manuscipt greatly.

My main comments for this version which need addressing:

Line 499: Bioinformatics method – in your results you have the numbers of reads and how many remained after QC. Please add to this section. A lot of reads didn’t pass the QC. Mothur should tell you how many samples remain after each QC step and chimera removal please include if possible. Comment in the discussion about the high error rate and relate to the literature on the platform and other studies. I believe these primers (Caporaso) when used for salmon pick up a lot of host DNA and so have a high error rate so could you comment on the source of the read loss (ie chimeras, read length, host DNA contamination etc).
Recommended vevision: Sentences in discussion discussing error rate where you already discuss low % of assigned species. More detail on QC.

I don’t agree with having diversity measures to order level I think it is too high. In the previous version there was inconsistency between figures and text showing different levels so understandably choosing one level is easiest and is clearer, but for these analysis the standard is OTU-level or genus maybe family-level. For example in the results it is stated ‘Alpha-diversity was similar among species and among types of samples’ I’m highlighting this as I think that is likely you are just not looking deep enough and missing something interesting in your study. Unfortunately many of your reads did not assign at genus/family level. As you state only 16% of your reads were assigned taxanomically.
Recommended vevision: Test alpha-diversity at OTU level.

The rarefaction results I would like to see at read-level to see the variation per group but also per sample. Especially given how many reads were removed through QC. From reading the manuscipt there’s no way of knowing if this is even per sample. The rarefaction at this level is crucial here. In fact you mention ‘samples having 10 times the number of different orders found in other samples from the same type. Something similar was found for the samples from each species, meaning that there was high heterogeneity in samples used in this study (rarefaction curves for each species and sample type’ Supplementary Figure 2)
Recommended vevision: Rarefaction curve with reads after QC or OTU level per sample.

For the rest of the analysis it is fine to be at order level. The description of interesting genera in the discussion reads well.

Other comments:
Line 9: Remove ‘is considered’, remove sensentence from partlly (as you go on to say the same thing in the next sentence).
Line 334: change ‘place where each individual was captured’ to ‘sampling location of each individual’.
Line 349: Ion torrent method – were samples pooled to equimolar concentrations? Perhaps add more brief but relevant details from the stated protocols.
Line 519: What R version?
Line 572: Keep sentence with how many OTUs found (is this 22,803?) and then how many taxonomically asigned remove the first part and put into methods as described above.

Figures:

Figure 1: I think Figure 1 and Figure 2b can go in supplements as one figure.
Recommended new figure 1a and b: A) Number of shared OTUs per sample type. I think this will be interesting as it tells us about the 84% of the community not tax assigned. Then Figure 2b) Number of shared orders per sample type.
Figure 5 should go to supplements as it is not needed. And the shark species cover parts of the plot the reader will want to see. See recommendation above for rarefaction curve.

Supplementary Figure 3: Labels on plot hard to read consider removing.

Image resolution quality may need to be improved for publication

Reviewer 3 ·

Basic reporting

The wording and the English of the manuscript has substantially imporved. It is very clear.

Experimental design

Correct

Validity of the findings

It is the first study assessing the microbiota of mucus and skin in 3 Elasmobranchs species, which is interesting to have a reference in future studies.

Additional comments

Both in the Abstract and Introduction authors speak about the behaviour of sharks during copulation, and the risk of infections. However, there is no discussion at all about the implications of the results obtained in this situation.

L21 and so on: If you decided to keep only the results on Orders, do not include the Genera.
L23-24: There is an error in the redaction of the sentences: “Bacillales, However…” There is no sense.
L24-28: Please rewrite those results since it is not clear what you mean.
L107: 90 % and 4 ºC, and be consistent all along the manuscript.
L196: Remove “and” after 28.
L238: due to a…
L231: Add: “or fish in general”
L410: Remove “and”
Fig. 8: were due to a…

---

## Round 0.4 · Minor Revisions

We are getting closer here. Can you please answer the last points raised by reviewer1? Also, to answer comment in line 155 it should be Qiime2 instead of Qiime.

·

Basic reporting

Covered in previous revisions.

Experimental design

Covered in previous revisions.

Validity of the findings

Covered in previous revisions.

Additional comments

Thanks to the authors for being very patient throughout this review process and for sticking with it - and gaining additional training in bioinformatics. I found the paper really interesting and a valuable contribution to the field so well done - only some minor comments below.

Note: These line numbers are from the tracked word document version:

Line 17: Define OTUs when you first mention in the text.
Line 24 + 25: Give the P value
Line 155: Is that the right reference for Qiime?
Line 177: Reference J Oksanen - ‎2019 for Vegan package – Vegan without all caps.
Line 212: change to occurrence
Line 340: change , to full stop
Line 350: Within text change “Alpha-diversity for OTUs” to Alpha-diversity at OTU-level” – same for others such as order level.
Line 453: Remove startling, perhaps start the sentence from the word “More”
Line 472: Do you show that in this study? Or you mean the reference you are citing?
Line 410:I would perhaps move around this paragraphs – you talk about pathogens 410-456, then have a paragraph on co-occurrence then pathogens again 472-506. So perhaps move the co-occurrence to fit in above when discussing the make up of the communities. It could be shortened to add further up.
Line 844: Is it meant to be (N#). In the figure legend?
Line 819: for OTUs is mentioned twice
Supplementary figure 3 – it’s unclear what the samples are as the reader won’t know your sample IDs. You will need to add a description to the legend. Or you can overlay a text box in powerpoint and re-save the image.

Reviewer 3 ·

Basic reporting

Very well written, everything is correct in this new version.
I have just a suggestion. L280: Change the comma to a full stop.

Experimental design

Everything is correct.

Validity of the findings

No additional comments

---

## Round 0.5 · Major Revisions

Hello. I have now gone over through this last version, and unfortunately, in my opinion, there are still some problems that need addressing :

1. Errors: there are some nomenclature errors, so please check all the taxa names.

2. Water sample diversity. It is extremely unusual, and you might need to explain it. If Actinomycetales is dominated by one or a few OTUs in all samples (an if the dominance is more important with samples with less DNA it could be a contaminant see :
DOI: 10.1128/mSystems.00290-19 and /doi.org/10.1186/s40168-018-0605-2. Also, it is very strange that those samples are all over in Figure 6. One really would expect them to be closer together.

3. Statistics: I cannot understand why you merged replicates in Figure 5 and in your PCA without those it is difficult to evaluate any of your conclusions

4. Discussion: You are extrapolating physiology and lifestyle at an order level (I.e. you are assuming that sulfur reduction is a characteristic of orders where it is not the case. You need to tone down the discussion. There are now many publications in microbiota of fish mucus. Maybe you should instead compare yours to results in bony fish rather than making these risky conclusions.

I have a list of modifications I would like you to make:

Title. Specify: Description of the microbiota in epidermal mucus and skin of the nurse shark (Ginglymostoma cirratum), the lemon shark Negaprion brevirostris) and the southern stingray (Hypanus americanus).
line 2 add doi number
line 23. Pseudoalteromonas is not in Vibrionales. I believe this is an error in the references used for taxonomy (i know that is a fact for the green genes database that is also based on SILVA). Also, with you description is very difficult to evaluate which version of the arb database and which classifier you used. Please add that information in the methods
line 47 viscous
line 49 compounds instead of substances
line 199 archaeal
line 138. add rRNA gene to 16S, Also, the link does not work, so I could not see the data
line 182. Does this include all samples including water?
line 206. Bacteria and Archaea are Domains not Kingdoms.
line 231. Merge with previous paragraph
line 236. Microbial, not microbiome. remove the term microbiome from the Manuscript
line 239 Figure 5 and 7 PCA. I don't understand why you merged the replicate individuals for each species. Can you explain? I would like to see a tree showing each of the replicates, and a mention in the text with the merged distances in the text
line 242. I do not see this in the figure, and I find hard to explain why the water samples are not more similar to each other and somehow more separated from the mucus samples. As someone who work with bacterioplankton for the last 30 years I also find very difficult to understand how Actinomycetales can make such a large proportion of reads in seawater, since I never seen this be the case in any of the datasets I ever worked with. This needs to be checked or at least discussed
line 254. see comment in 239. There's gotta be a reasoning for "merging" the replicates
line 259, delete "at this time"
line 264. Furobacteriales is not a prokaryotic order
line 290 can you refer to the results you use to make this conclusion?
line 302. Figure 6 seems to contradict this statement.
line 378-393. You are extrapolating results based on orders to very different animals. I ask you to remove this entire paragraph
line 397 I do not believe there is a bacterial order called Synachococcales neither Halanaenobiales,
line 398. I do not see the term desufurating used widely. Plese use "sulfate reducing"
line 399. orders
line 402. As far as I know Sulfate reduction is not a major characteristic of all these orders, so attributing it to the cooccurrence of these is over speculation. moreover Synechococcales is an oxygenic phototroph. I ask you to remove this from the discussion.
line 406. The fact that two species co-occur does not say much about their occurrence. The presence of fusobacteria confirms previous results e.g. https://doi.org/10.1007/s00248-020-01484-y https://doi.org/10.1093/femsec/fix051
line 409. This study did not study the role of mucus or its bacteria. You could use. "Here we show the presence of bacterial orders that could be involves on pathogenicity"
line 457 orders
line 458. Unless you can justify it with references remove the section related to desulfuration.
Figure 5. It is a Chord diagram

·

Basic reporting

Addressed in previous revisions.

Experimental design

Addressed in previous revisions.

Validity of the findings

Addressed in previous revisions.

Additional comments

Thanks to the authors for making all these suggested changes through previous drafts. The manuscript is much improved through these revisions and represents a valuable piece of work for the research community.

---

## Round 0.6 · accepted · Accept

I have been asked to handle this resubmission after the previous editor resigned from this manuscript. I am sorry for the delay, but it took me a while to come up to speed, and I have now read through all the previous reviews and your responses. The referees are well-qualified to evaluate this work and have provided 4 rounds of detailed review in response to which you have extensively revised your manuscript to incorporate their feedback. Ultimately, you have made all the changes requested by the referees and they have recommended the revised manuscript be accepted for publication. I note that the addition of text explaining your PCR negative controls to the body of the manuscript will add confidence among readers that the high presence of Actinomycetales is not the result of PCR contamination. Beyond some skepticism about that unexpected result, there is nothing I see in any of the referee comments that is justification to prevent this work from being published. Your 5 rounds of revision have addressed every serious concern raised in the review process and I see no reason to delay this process any further. Therefore, I am happy to move it forward into production.